# Gram-Negative ESKAPE Bacteria Surveillance in COVID-19 Pandemic Exposes High-Risk Sequence Types of *Acinetobacter baumannii* MDR in a Tertiary Care Hospital

**DOI:** 10.3390/pathogens13010050

**Published:** 2024-01-04

**Authors:** Mónica Alethia Cureño-Díaz, Estibeyesbo Said Plascencia-Nieto, Miguel Ángel Loyola-Cruz, Clemente Cruz-Cruz, Andres Emmanuel Nolasco-Rojas, Emilio Mariano Durán-Manuel, Gabriela Ibáñez-Cervantes, Erika Gómez-Zamora, María Concepción Tamayo-Ordóñez, Yahaira de Jesús Tamayo-Ordóñez, Claudia Camelia Calzada-Mendoza, Juan Manuel Bello-López

**Affiliations:** 1Hospital Juárez de México, Mexico City 07760, Mexico; 2Facultad de Ciencias de la Salud, Doctorado en Ciencias de la Salud, Universidad Anáhuac, Naucalpan de Juárez 52786, Mexico; 3Sección de Estudios de Posgrado e Investigación, Escuela Superior de Medicina, Instituto Politécnico Nacional, Mexico City 11340, Mexico; 4Laboratorio de Ingeniería Genética, Departamento de Biotecnología, Facultad de Ciencias Químicas, Universidad Autónoma de Coahuila, Coahuila 25280, Mexico; 5Laboratorio de Biotecnología Ambiental, Centro de Biotecnología Genómica, Instituto Politécnico Nacional, Reynosa 88710, Mexico

**Keywords:** ESKAPE bacteria, surveillance, hospital infection, antimicrobial resistance, sequence type, *Acinetobacter baumannii*, COVID-19 pandemic

## Abstract

The interruption of bacteriological surveillance due to the COVID-19 pandemic brought serious consequences, such as the collapse of health systems and the possible increase in antimicrobial resistance. Therefore, it is necessary to know the rate of resistance and its associated mechanisms in bacteria causing hospital infections during the pandemic. The aim of this work was to show the phenotypic and molecular characteristics of antimicrobial resistance in ESKAPE bacteria in a Mexican tertiary care hospital in the second and third years of the pandemic. For this purpose, during 2021 and 2022, two hundred unduplicated strains of the ESKAPE group (*Escherichia coli*, *Klebsiella pneumoniae*, *Pseudomonas aeruginosa*, and *Acinetobacter baumannii*) were collected from various clinical sources and categorized by resistance according to the CLSI. An analysis of variance (ANOVA) complemented by the Tukey test was performed to search for changes in antimicrobial susceptibility profiles during the study period. Finally, the mechanisms of resistance involved in carbapenem resistance were analyzed, and the search for efflux pumps and high-risk sequence types in *A. baumannii* was performed by multilocus analysis (MLST). The results showed no changes in *K. pneumoniae* resistance during the period analyzed. Decreases in quinolone resistance were identified in *E. coli* (*p =* 0.039) and *P. aeruginosa* (*p =* 0.03). Interestingly, *A. baumannii* showed increases in resistance to penicillins (*p =* 0.004), aminoglycosides (*p <* 0.001, *p =* 0.027), carbapenems (*p =* 0.027), and folate inhibitors (*p =* 0.001). Several genes involved in carbapenem resistance were identified (*bla_NDM_*, *bla_VIM_*, *bla_OXA_*, *bla_KPC_*, *bla_OXA-40_*, and *bla_OXA-48_*) with a predominance of *bla_OXA-40_* and the *adeABCRS* efflux pump in *A. baumannii.* Finally, MLST analysis revealed the presence of globally distributed sequence types (ST369 and ST758) related to hospital outbreaks in other parts of the world. The results presented demonstrate that the ESKAPE group has played an important role during the COVID-19 pandemic as nosocomial antibiotic-resistant pathogens and in particular *A. baumannii* MDR as a potential reservoir of resistance genes. The implications of the increases in antimicrobial resistance in pathogens of the ESKAPE group and mainly in *A. baumannii* during the COVID-19 pandemic are analyzed and discussed.

## 1. Introduction

In 2008, Rice first coined the acronym ESKAPE for a group of bacteria that in addition to being multidrug-resistant by “escaping antimicrobial therapy”, it was observed that it was a group of bacteria prevalent as causative agents of healthcare-associated infections (HAIs). This observation, which was not so obvious to many, served to focus the attention of several hospital centers around the world on this group of microorganisms, calling them “ESKAPE pathogens” [1,2,3,4,5,6,7]. During the COVID-19 pandemic, this bacterial group was one of the main agents causing co-infections in critically ill patients, where ventilator-associated pneumonia (VAP) was the most important HAI due to the drug resistance identified in these isolates, as well as a generator of hospital outbreaks [8,9,10]. It has been shown that in patients with any HAI caused by antibiotic-resistant ESKAPE bacteria, morbidity and mortality and hospital care costs are dramatically increased [11,12,13]. For example, Sosa-Hernandez et al. (2019) showed that VAP caused by MDR bacteria (mainly the ESKAPE group) confers nine times the risk of increasing the costs of care above the expected average [14]. In other regions of the world, the negative impact in terms of costs of HAIs has been demonstrated, where VAP is considered an infectious complication that dramatically increases care costs and mortality [15,16,17]. Therefore, hospital epidemiological surveillance of their incidence, resistance, and their circulating high-risk mechanisms and sequence types should be a priority activity. Classically, the acronym ESKAPE is composed of bacteria such as *Enterococcus faecium*, *Staphylococcus aureus*, *Klebsiella pneumoniae*, *Acinetobacter baumannii*, *Pseudomonas aeruginosa*, and species of the genus *Enterobacter* [18]. However, with the advances in the epidemiological surveillance of in-hospital infectious agents, it has been shown that this acronym can change according to the nosocomial environment since it has been observed that, between each hospital, the diversity of circulating microorganisms can be radically different. Peterson (2009) suggested the inclusion of the anaerobic bacterium *Clostridium difficile* (currently known as *Clostridioides difficile*) in the ESKAPE group, justifying its inclusion, since its worldwide identification as a causative agent of HAIs was increasing [19,20,21]. In Mexico, this microorganism has been recognized as a causative agent of hospital outbreaks in critical patients and has led to the implementation of epidemiological surveillance programs to contain future outbreaks [21].

Currently, *C. difficile* continues to be one of the main “fastidious” anaerobic sporulating microorganisms worldwide, and its inclusion in the ESKAPE group remains controversial. With the modification of this acronym to ESCAPE, all *Enterobacteriaceae*, recently renamed *Enterobacterales*, had to be included as a bacterial group of medical importance and associated with various HAIs. Another example of a bacterial pathogen that has recently been recognized as a potential ESKAPE member is *Stenotrophomonas maltophilia,* which has been the causative agent of in-hospital outbreaks among critically ill patients [22,23,24]. In Mexico, we have locally reported the members of the ESKAPE group resistant to antibiotics (mainly Gram-negative bacteria) circulating in pre-pandemic and pandemic times in patients of Hospital Juárez de México (HJM) [7,14], as well as the consequences of their dispersion on inert surfaces and critical devices, such as *A. baumannii* [5,25,26,27,28]. Therefore, microbiological evidence from our hospital has shown that only some Gram-negative bacteria (*A. baumannii*, *P. aeruginosa*, and *Enterobacterales*) have played an important role as causal agents of HAIs and nosocomial contamination. Recent theories indicate that the COVID-19 pandemic brought with it several consequences, among which is the increase in antimicrobial resistance in the ESKAPE group, both events being considered “syndemic” because of their close association and involvement in COVID-19 patients [29,30]. According to the Pan American Health Organization (PAHO), the syndemic is reinforced due to the irrational and indiscriminate use of antibiotics to empirically treat patients with COVID-19 of confirmed infections, not of bacterial origin. For example, in HJM, empirical therapy based on meropenem, imipenem, and piperacillin/tazobactam was used [7], so we can speculate that this type of non-directed treatment, together with shortage of antibiotics and poor clinical practices in the management of COVID-19 patients, among others, favored the increase in antimicrobial resistance. From a genetic point of view, it is not difficult to speculate that the genomic plasticity of the ESKAPE group favors the acquisition of resistance mechanisms and consequently increases antimicrobial resistance rates during and after the pandemic. Therefore, the aim of this work was to identify the possible rates of increase in antimicrobial resistance in Gram-negative ESKAPE pathogens causing HAIs in HJM during the second and third years of the COVID-19 pandemic, the genetic mechanisms involved in carbapenem resistance, with an emphasis on high-risk sequence types of *A. baumannii* global distribution. The implications for increases in antimicrobial resistance in Gram-negative ESKAPE pathogens during the COVID-19 pandemic are analyzed and discussed.

## 2. Materials and Methods

### 2.1. Origin of the Gram-Negative ESKAPE Strains, Bacterial Identification, and Controls

The Gram-negative ESKAPE strains (*Escherichia coli*, *Klebsiella pneumoniae*, *Pseudomonas aeruginosa*, and *Acinetobacter baumannii*) were isolated from the non-repeat patients possessing confirmed HAIs (without previous antimicrobial treatment) from HJM, during 2021 (*n =* 114) and 2022 (*n =* 86). These periods corresponded to the second/third and fourth waves of the COVID-19 pandemic for the years 2021 and 2022, respectively. Because Gram-negative bacteria from the ESKAPE group have been the most prevalent as causal agents of HAIs in HJM, only these were included in the study. The Gram-negative ESKAPE strains were isolated from different clinical sources: urine (urinary tract infections), wounds (purulent infections), lung (ventilator-associated pneumonias from COVID-19 patients), and blood (sepsis). The identification to genus and species level was performed using the automated system Vitek 2-XL (bioMériux, Durham, NC, USA) according to the manufacturer’s protocol. *Klebsiella pneumoniae* 466 (*bla_NDM-1_*), *P. aeruginosa* PA-11 (*bla_VIM_*), *K. pneumoniae* ATTC BAA-1705 (*bla_KPC_*), and *K. pneumoniae* BAA-2524 (*bla_OXA-48_*) strains were used as positive controls, and *E. coli* J53-1 as negative control [7,25]. Only for the detection of the *bla_OXA-40_* and *bla_OXA-23_* genes, the identity of the amplicons was sequenced by the Biology Institute of *Universidad Nacional Autónoma de México* (UNAM) using a DNA Analyzer 3730xL (Applied Biosystems, Forrest City, CA, USA). Nucleotide sequences were compared with the nucleotide sequence database (GenBank) by means of the Blast algorithm (http://blast.ncbi.nlm.nih.gov (accessed on 15 October 2023)), using parameters of coverage (>80%) and identity (90%).

### 2.2. MDR, XDR, and PDR Classification of Gram-Negative ESKAPE Bacteria

The antimicrobial resistance profile was determined through the guidelines established by CLSI (2022) [31], and the classification of MDR (multidrug-resistant), XDR (extensively drug-resistant), and PDR (pandrug-resistant) Gram-negative ESKAPE bacteria was conducted according to “Latin American consensus to define, categorize and notify multidrug-resistant pathogens, with widespread resistance or pan-resistant pathogens” [32]. *Pseudomonas aeruginosa* ATCC 27853 and *E. coli* ATCC 25922 were used as controls. Results were inferred as susceptible, intermediate, or resistant by measuring the diameter of the inhibition zone. The frequency of antibiotic resistance was calculated and represented in percentages (%), and MDR, XDR, and PDR classification was performed through heat maps. The visualization of the distribution of MDR and XDR phenotypes of Gram-negative ESKAPE isolates and isolate sources was analyzed using ShinyCircos software (https://github.com/venyao/shinyCircos (accessed on 17 October 2023)) according to Yu et al. (2018) [33].

### 2.3. Carbapenemase Production and Their Relationship with Genotypes

#### 2.3.1. Carbapenemase Detection by mCIM Assay

Gram-negative ESKAPE bacteria strains were subjected to the modified carbapenem inactivation method (mCIM) according to Pierce et al. (2017) [34]. In brief, two 1 μL loopfuls of Gram-negative ESKAPE bacteria from an overnight culture were emulsified in 2 mL trypticase soy broth (TSB). Subsequently, a 10 μg MEM disk (BD, Brea, CA, USA) was immersed in each suspension and incubated at 37 °C for 4 h. Furthermore, a Mueller–Hinton (MH) was massive plated with *E. coli* ATCC 25922 (MEM^S^) suspension adjusted to 0.5 McFarland nephelometer. Finally, MEM disks were removed from the bacterial suspension and were deposited on MH plates with the indicator MEM^S^ strain. MH plates were incubated at 37 °C for 18–24 h and the zones of inhibition were measured according to the routine disk diffusion method. *Klebsiella pneumoniae* carrying *bla_NDM-1_* gene was used as the positive control.

#### 2.3.2. Screening to Confirm Carbapenemases Production in Gram-Negative ESKAPE Bacteria

Regarding the genetic background in Gram-negative ESKAPE bacteria that confer resistance to β-lactams (carbapenems), end-point PCR assays were performed to detect metallo-β-lactamases (*bla_NDM_*, *bla_VIM_*, and *bla_IMP_*) and serine β-lactamases (*bla_KPC_*_,_
*bla_OXA-48_*, *bla_OXA-23_*, and *bla_OXA-40_*) genes, using the primers previously described. Amplicons were run in 1 × TBE buffer (pH 8.3), separated via horizontal electrophoresis in 2.0% agarose gels, visualized, compared with an appropriate molecular weight marker, and photographed under UV illumination. The primers used for this purpose are shown in Table 1.

### 2.4. adeABC Operon and Regulator Gene Detection adeRS in A. baumannii

Full operon *adeABC* encoding efflux pumps and their regulator genes *adeRS* were amplified by end-point PCR in *A. baumannii* strains according to Durán-Manuel et al. (2021) [5]. Detection was performed using the amplification strategy of conserved genes as follows: an initial PCR reaction was performed to amplify the *adeA* gene encoding a protein forming a dimeric complex that anchors in the periplasmic region of the cell. Once a positive amplification to the first molecular target was performed, a second reaction to amplify the *adeB* gene (encoding an intermembrane protein) was carried out. 

Finally, a third reaction was performed to amplify the *adeC* gene (encoding an extramembrane protein). Additionally, the *adeR* and *adeS* genes encoding a regulator protein and activating protein kinase, respectively, were amplified. Amplicons were run and separated using horizontal electrophoresis, compared with a marker of the appropriate molecular weight, and photographed under UV illumination. The sequences of primers used for *adeABCRS* operon amplification are shown in Table 1. 

### 2.5. Detection of High-Risk Sequence Types in Acinetobacter baumannii

Due to the high frequency of *A. baumannii* MDR strains identified in this study, together with the identification of significant changes in antimicrobial resistance during the pandemic and under the premise that this microorganism is the main causative agent of outbreaks and nosocomial problems in HJM and other parts of the world, a search for high-risk sequence types of global distribution was performed by multilocus analysis (MLST) for this ESKAPE member. For this purpose, the sequence of seven constitutive genes (*gltA*, *gyrB*, *gdhB*, *recA*, *cpn60*, *gpi*, and *rpoD*) previously amplified by PCR according to Bartual et al. (2005) [39] (Table 1) was analyzed. These constitutive genes were proposed under the Oxford University scheme. The analysis was performed using the online Multi Locus Sequence Typing software available at https://pubmlst.org/mlst/ (accessed on 21 October 2023).

### 2.6. Statistical Analysis

An analysis of variance (ANOVA) was performed to determine the variation between the means of the antimicrobial susceptibility profile (for each antibiotic) of the Gram-negative ESKAPE strains isolated during 2021 and 2022. Additionally, Tukey’s test was performed to determine whether there was a significant difference (*p* = 0.05) between the means of the data obtained from the susceptibility profiles (for each antibiotic). For this purpose, qualitative variables were assigned a label value of 1, 2, and 3 for resistant, intermediate, and sensitive profiles, respectively. SPSS v.27.0.1.0 and XLSTAT 2023 statistical software was used for the analysis and graphical representation.

## 3. Results

### 3.1. Gram-Negative ESKAPE Bacterial Population Included in This Study

A total of 200 strains belonging to the Gram-negative ESKAPE group causing nosocomial infections during 2021 (*n =* 114/57%) and 2022 (*n =* 86/43%) were included in this study. The prevalent Gram-negative ESKAPE member was *E. coli* with a frequency of 49%, followed by *P. aeruginosa* with 24.5%. For *K. pneumoniae* and *A. baumannii*, the least prevalent members of the Gram-negative ESKAPE group, showed frequencies of 13.5 and 13%, respectively. Table 2 shows the frequency distribution by year (2021 and 2022) of Gram-negative ESKAPE group strains analyzed in this study from patients of HJM.

### 3.2. Clinical Origin of Gram-Negative ESKAPE Bacteria

The analysis of the origin of the isolation of Gram-negative ESKAPE members showed that VAP and UTIs were the most prevalent HAIs with frequencies of 39.5% and 38%, respectively. In relation to wound and blood infection cases, frequencies of 10 and 12.5%, respectively, were identified. Table 3 shows the distribution of HAI cases in the isolation of bacteria from the Gram-negative ESKAPE group during the years 2021 and 2022 from patients of HJM.

### 3.3. MDR, XDR, and PDR Classification of Gram-Negative ESKAPE Bacteria

#### 3.3.1. Antimicrobial Resistance Gram-Negative ESKAPE Bacteria (*Enterobacterales*)

To know the antimicrobial resistance patterns of the Gram-negative ESKAPE population during 2021 and 2022, and the possible changes in resistance rates, heat maps of antimicrobial resistance profiles against tested antibiotics were generated. Initially, the first classification of Gram-negative ESKAPE bacteria by fermenters (*E. coli* and *K. pneumoniae*) and non-lactose fermenters (*P. aeruginosa* and *A. baumannii*) by year and source of isolation was performed (Figure 1 and Figure 2). As a second step, only with the resistance profiles, Gram-negative ESKAPE isolates were classified as MDR, XDR, and PDR according to isolation source (Figure 3). 

The results of this analysis revealed that for *E. coli*, antibiotics of the penicillin family showed the highest resistance (AMP/82, SAM/55, and AMC/66%), followed by cephalosporins (FEP/66, CAZ/67, and CRO/72%), quinolones (CIP/84%) and folate metabolism inhibitors (SXT/62%). Interestingly, TZP, FOX, aminoglycosides, carbapenems, and lipoglycopeptides were the antibiotics where isolates showed the least resistance (Figure 1A). For the second fermenting ESKAPE member (*K. pneumoniae*), AMP was the only antibiotic with lower antimicrobial activity, since the population analyzed showed 100% resistance. Similarly, FOX (33%), aminoglycosides (AN/7 and GM/30), carbapenems (MEM/22 and IPM/11%), and lipoglycopeptides (0%) were the antibiotics where the population showed lower resistance compared to *E. coli* (Figure 1B).

It was identified that more than 50% of the *E. coli* and *K. pneumoniae* populations were classified as MDR strains, with 53% and 59%, respectively. Figure 1 shows the heat map of the antimicrobial resistance profiles of the *Enterobacteriaceae* group (*E. coli* and *K. pneumoniae*) of the Gram-negative ESKAPE group of HJM, showing that MDR isolates of *E. coli* (*n =* 30) and *K. pneumoniae* (*n =* 6) were predominantly urinary and pulmonary, respectively, but were identified in the rest of the infectious processes studied (Figure 3).

#### 3.3.2. Antimicrobial Resistance Gram-Negative ESKAPE Bacteria (Non-Lactose Fermenting)

For the *P. aeruginosa* group, levels below 50% resistance were identified for all antibiotics tested. Resistance analysis revealed the presence of MDR and XDR strains in 16.3% and 22.2%, respectively, with pulmonary and urinary predominance (Figure 2 and Figure 3). The susceptibility profiles showed that lipoglycopeptides remain the best therapeutic option for eradication of this MDR and XDR bacteria (Figure 2 (left)). Finally, even though the *A. baumannii* population was the smallest, high rates of resistance were identified against the fourteen antibiotics tested. These rates allowed 92.3% of the population to be classified as MDR. Rates of 7.7% of this same microbial group were identified as “sensitive”. A large proportion of the MDR isolates from this bacterial group showed pulmonary predominance. Figure 2 (right) shows the heat map of the antimicrobial resistance profiles of the non-fermentative group (*P. aeruginosa* and *A. baumannii*) of the Gram-negative ESKAPE group at HJM. 

#### 3.3.3. Changes in Antimicrobial Resistance in *Enterobacterales* during 2021 and 2022

To know the possible variations in the increase or decrease in antimicrobial resistance in the study period, an analysis of variance coupled with Tukey’s test was performed for the resistance phenotypes in *Enterobacterales*. The results of the analysis revealed that there was no significant difference (*p =* 0.05) between the means of the results of the resistant phenotypes obtained from the *E. coli* profiles in 14 of the 15 antibiotics tested. A significant difference in resistance (with a downward trend in 2022) was only identified for ciprofloxacin (*p =* 0.039). Finally, no significant variation was detected in the *K. pneumoniae* population, and the global identification of MDR isolates in the *Enterobacterales*. Figure 4 shows the analysis of variance and Tukey’s test for the identification of changes in the resistant phenotypes of the group of *Enterobacterales* of Gram-negative ESKAPE bacteria from HJM.

#### 3.3.4. Changes in Antimicrobial Resistance in Non-Fermenters during 2021 and 2022

The *P. aeruginosa* population showed no variation in the resistant phenotype for all antibiotics tested, except for ciprofloxacin, which showed a significant downward change in 2022 (*p =* 0.003). Interestingly, for the *A. baumannii* population, a member of the Gram-negative ESKAPE group of major importance for our hospital, significant differences (upward in 2022) were identified for five antibiotics distributed in four different families (penicillins, aminoglycosides, carbapenems, and folate metabolism inhibitors/sulfonamides). For SAM, a value of *p =* 0.004 was identified, as well as AN (*p* ≤ 0.001), GM (*p =* 0.027), MEM and IPM (*p =* 0.027), and STX (*p* ≤ 0.001). Finally, a significant increase in the number of MDR isolates was identified in 2022 (*p =* 0.027). Figure 5 shows the analysis of variance and Tukey’s test for the identification of changes in the resistant phenotypes of the non-fermenting group of Gram-negative ESKAPE bacteria from HJM.

### 3.4. Carbapenemase Production and Their Relationship with Genotypes

Phenotypically, a total of 68 (34%) isolates from the Gram-negative ESKAPE group were resistant to carbapenemases in the first stage (by CLSI), where *A. baumannii* isolates were the prevalent member (35.4%), followed by *P. aeruginosa*, *E. coli*, and *K. pneumoniae.* Conversely, the results of the confirmatory test by mCIM assay showed that only 31 isolates representing 31% of the Gram-negative ESKAPE isolate population were true carbapenemase producers. The predominant producing microorganism was *A. baumannii*, where serine β-lactamase (*bla_OXA-40_*) was involved as the mechanism of carbapenem resistance. The second group of carbapenemase producers was represented by *E. coli*, carrying metallo-β-lactamases (*bla_NDM_*, *bla_VIM_*, and *bla_KPC_*) and serine β-lactamases (*bla_OXA-48_*) as resistance mechanisms. The coexistence of the metallo-β-lactamases *bla_NDM_* and *bla_VIM_* was identified in an isolate of *K. pneumoniae* of pulmonary origin. Table 4 shows the results of the stages in the detection of carbapenems and the associated resistance mechanisms in the members of the Gram-negative ESKAPE group of HJM. In contrast, all carbapenem-resistant isolates of *A. baumannii* (100%) were able to identify the complete *adeABC* operon and the regulatory operon *adeRS.*

### 3.5. Detection of High-Risk Sequence Types in Acinetobacter baumannii

The results of the MLST analysis proposed by the Oxford scheme for *A. baumannii MDR* isolates (*n* = 24) revealed the presence of three high-risk sequence types of global distribution represented by ST369 (*n* = 14/58.4%) and ST758 (*n* = 7/29.1%), and one of local distribution, ST1679 (*n* = 3/12.5%). According to the identification of the high-risk sequence types during the study period, no outbreaks were identified among patients (Figure 2 (right)). All the high-risk clones identified carried the gene encoding *bla_OXA-40_* and the efflux pump *adeABCRS*.

## 4. Discussion

The worldwide interruption of epidemiological surveillance in infection prevention and control practices, as well as the diversion of human and financial resources for the control of the COVID-19 pandemic, brought with it several consequences, including the collapse of health systems and, in bacteriological terms, the possible increase in antimicrobial resistance. This highlights the need to understand resistance rates and their associated mechanisms in bacteria causing hospital-acquired infections. Therefore, the aim of the present work was to demonstrate the phenotypic and molecular characteristics of antimicrobial resistance in the main members of the ESKAPE group (Gram-negative) causing infections in HJM patients during the second and third years of the COVID-19 pandemic. This was performed to show the possible increase in antimicrobial drug resistance during the pandemic period. The results obtained showed that the highest isolation rates of microorganisms from the Gram-negative ESKAPE group were *E. coli* related to cases of urinary tract infections, together with *K. pneumoniae*, both located in the group of *Enterobacteriaceae* (Table 2 and Figure 1). Conversely, they were also detected as causative agents of VAP in COVID-19 patients, wound infections, and septic processes. *Enterobacterales* have been recognized as important pathogens in urinary tract infections, since they are part of the intestinal microbiota of warm-blooded animals; however, since they are also part of the nosocomial reservoir, they can cause infections in sites anatomically different from the urinary tract, such as those identified in this and other studies [40,41,42,43]. It is important to note that statistical analyses of antimicrobial drug resistance for the period analyzed showed no significant upward changes in these two enterobacteria; on the contrary, a decrease in the rate of resistance to a quinolone (ciprofloxacin, *p =* 0.039) was identified. Interestingly, this finding is opposite to that reported by Wardoyo et al. (2021), where they reported a significant increase in resistance to ofloxacin in a population of 210 *E. coli* isolates of diverse clinical origins (*p <* 0.05) [44]. Alternatively, they observed a decrease in resistance to piperacillin (*p =* 0.012), amoxicillin (*p =* 0.002), cefadroxil (*p =* 0.036), and ampicillin (*p =* 0.036). It is not difficult to speculate that the differences in resistance patterns in these microorganisms compared to our work may be related to the different nosocomial settings and to the epidemiological surveillance that may or may not be successful, as well as the appropriate management of antimicrobials. 

Alternatively, there is a possibility that because there was a smaller population in terms of isolation for *K. pneumoniae* (*n* = 27) compared to *E. coli* (*n* = 98), there was an incorrect assessment of the resistance rates identified; however, the statistical analysis supports the absence of variation in resistance in *K. pneumoniae*. Other reports have shown an increase for antibiotics in the carbapenem family, antibiotics used as a last resort (prior to colistin) for the treatment of MDR infections. Chatterjee et al. (2023), through a comparative study between pre-pandemic and pandemic periods in an Indian hospital, demonstrated an increase in carbapenem resistance in *K. pneumoniae* and *P. aeruginosa* from 23 to 41% between the two periods analyzed [45]. Theories such as the increase in patients with comorbidities in the COVID-19 period, and the reduction in microbiological surveillance (reporting of microbiological contamination of surfaces, critical, and semi-critical devices to antimicrobial control committees and infection control committees, among others), increases in the population treated in hospitals, and the unnecessary, irrational, and improper use of antibiotics were recognized as potential causes for the increase in drug resistance in these pathogens. Regarding non-fermentative Gram-negative ESKAPE group bacteria and HAIs, the results showed that during the period analyzed, VAP in COVID-19 patients was one of the most prevalent infections, where *P. aeruginosa* was the main microorganism involved, followed by *A. baumannii* (Figure 2). This observation has been previously identified by our group in the pre-pandemic period in this same hospital, where Sosa-Hernandez et al. (2019) demonstrated that *P. aeruginosa* and *A. baumannii* were the main pathogens of VAP, with rates of 16.7% and 47.9%, respectively in 48 patients over a whole year of analysis [14]. 

In contrast, during the first year of the pandemic, isolation rates increased dramatically in VAP cases in COVID-19 patients, from 20% (*n* = 19) and 15% (*n* = 14) for *P. aeruginosa* and *A. baumannii* in 96 cases in only two months of the first pandemic wave [7]. From this work it was concluded that *P. aeruginosa* (carrier of the gene *bla_VIM_*) and *A. baumannii* (carrier of the genes *bla_OXA-23_* and *bla_VIM_*) continue to be the main causative agents of HAIs in pandemics. Finally, in the present work, it was observed that during the second and third years of the pandemic, rates continued to increase, with 25 and 39% of patients having *A. baumannii* and *P. aeruginosa*, respectively, as the causative agent. Such observations of gradually increasing isolation rates of *P. aeruginosa* and *A. baumannii* in other countries have been reported before and after the COVID-19 pandemic, as well as for other microbiological agents, such as *E. coli* and *K. pneumoniae* [46,47,48]. 

Among the findings of our work, it was observed that the therapeutic options for the treatment of VAP were based on colistin and amikacin for *P. aeruginosa* and only colistin for *A. baumannii*, since the frequency of carbapenemase production was high for these microorganisms. It is important to note that even though *P. aeruginosa* showed high susceptibility to various antimicrobials, MDR and XDR isolates were identified (Figure 3), which have also been identified in critically ill patients [49]. Regarding the overall analysis of antimicrobial resistance in non-fermenting Gram-negative ESKAPE isolates, it was observed that resistance to ciprofloxacin in *P. aeruginosa* showed significant downward difference (*p =* 0.03), while resistance to the other antibiotics showed no statistically significant difference. These data may be epidemiologically encouraging in comparison to work where resistance rates have been found to be increased where carbapenems are included [46,47,48]. Finally, the only non-fermenting Gram-negative ESKAPE member that showed upward changes in antimicrobial drug resistance profiles was *A. baumannii*. Interestingly, statistically significant changes in resistance to β-lactam antibiotics (ampicillin/sulbactam), aminoglycosides (amikacin and gentamicin), carbapenems (meropenem and imipenem) and a folate metabolism inhibitor (trimethoprim/sulfamethoxazole) were observed in this member. These resistances are typically conferred by the presence of mobile genetic elements, where conjugative plasmids carrying genes coding for carbapenemases play an important role in the dissemination of carbapenem resistance [50,51,52]. 

Alternatively, integrons are among the genetic elements that have gained relevance in recent decades as carriers of antimicrobial resistance cassettes of the families and that have been identified in *A. baumannii* [53,54]. Against this background, we searched for the genetic mechanisms associated with carbapenemase synthesis in the Gram-negative ESKAPE bacterial population. As reported by Loyola-Cruz et al. (2023b), the prevalent genetic marker in *A. baumannii* during the first year of the COVID-19 pandemic was a serine β-lactamase encoded in the *bla_OXA-23_*_,_ gene, contrasted with the findings of the second and third years of the pandemic, where this genetic marker underwent replacement by the serine β-lactamase marker *bla_OXA-40_* [7]. This suggests the emergence of new carbapenem-resistant strains of *A. baumannii* endowed with other resistance mechanisms, such as efflux pumps, as previously detected, consequently conferring resistance to a wide variety of antibiotics, including those where a significant increase was identified [5,7]. In this context, carbapenem resistance, in addition to being associated with the presence of carbapenemases, efflux pumps have been shown to be associated with multidrug resistance in *P. aeruginosa* and *A. baumannii* [55,56]. Therefore, we do not rule out the possibility that carbapenem resistance in those isolates of *A. baumannii* that did not carry genes coding for carbapenemases possessed efflux pumps involved in the resistance. 

According to the PAHO, during the pandemic, antimicrobial resistance was based on the irrational and indiscriminate use of antibiotics to empirically treat COVID-19 patients. For example, in HJM, empirical therapy based on meropenem, imipenem, and piperacillin/tazobactam was used [7]; therefore, we can speculate that this type of non-directed treatment, together with the shortage of antibiotics and the relaxation of good clinical practices, favored the increase in antimicrobial resistance in the *A. baumannii* population, since the resistance markers identified were directly related to the antibiotics used during the pandemic period. The evidence generated reinforces the role of *A. baumannii* as one of the main pathogens of the Gram-negative ESKAPE group in HJM; for example, in the work published by Durán-Manuel et al. (2021), where through the molecular analysis of intergenic spacer regions, the clonal dispersion of this microorganism was demonstrated for the first time, and that it also carried a resistance mechanism conferred by an efflux pump (*adeABCRS*) [5]. This resistance mechanism in this pathogen confers resistance to a wide range of antibiotics including carbapenems. The evidence presented by Durán-Manuel et al., clearly shows that medical devices used for COVID-19 patients are vehicles of pathogen transmission. Conversely, this microorganism has also been recognized as one of the main members of the Gram-negative ESKAPE group implicated in intrahospital outbreaks due to its permanence as a contaminating bacterium, due to its high adherence in the inhalation and exhalation branches of mechanical ventilators of COVID-19 patients. Cureño-Díaz et al. (2021) were able to identify and contain outbreaks among COVID-19 patients by modifying the cleaning and disinfection procedures of mechanical ventilators, critical devices used in the respiratory support of COVID-19 patients [27]. 

Another report on the spread of *A. baumannii* MDR is that of Shafigh et al. (2022), where they demonstrated the spread of this pathogen in seven COVID-19 patients. Analysis of the cases showed that even with colistin and ampicillin/sulbactam treatment, mortality rates were 100% [57]. This reflects the direct impact on morbidity and mortality in patient populations infected with MDR bacteria, such as those identified in the present study (Figure 3). Epidemiological knowledge of the dissemination of high-risk clones of *A. baumannii* goes beyond knowing the local epidemiological behavior of these isolates, since it has been shown that, using molecular typing techniques, it is also possible to know the genetic relationship of isolates that may be geographically distant, and due to their multidrug-resistant genetic background, they have been the cause of in-hospital outbreaks throughout the world. Therefore, the final purpose of this work was the investigation of high-risk clones and their relationship with other related clones in hospital-acquired dissemination events in other regions of the world. As can be seen in the results, the genomic diversity of *A. baumannii*, three STs were identified by multilocus analysis (MLST): 369, 758, and 1679. Interestingly, ST369 and ST758 are characterized as MDR, which have led to epidemic outbreaks and have been shown to play an important role in the increasing dissemination of antibiotic resistance mechanisms, even more so in the COVID-19 pandemic. In a study by Hwang et al. (2021), genetic analysis using various molecular and bioinformatics tools was performed on eleven carbapenem-resistant *A. baumannii* isolates associated with a hospital outbreak in Korea [58]. 

This study revealed the presence of high-risk STs, including the one identified in this study (ST369). Locally in our country, this clone has been identified in *A. baumannii* MDR causing an in-hospital outbreak [38]. The second ST (758) is characterized as MDR and is related to infectious processes in immunologically compromised patients. In a study involving several Colombian hospitals, 32 patients were analyzed, 13 of whom presented various extrapulmonary infections with *A. baumannii*. MLST analysis revealed circulating ST758 together with ST229 with *bla_OXA-23_* genetic markers [59,60]. Lastly, to our knowledge, ST1679 has not been formally reported in scientific manuscripts; however, there are reports in international databases indicating that this MDR clone was previously reported at the State Key Laboratory of Trauma, Burns and Combined Injury, Chongqing Key Laboratory for Proteomics Disease, Institute of Burn Research, Southwest Hospital, the Third Military Medical University [61]. 

Undoubtedly, MLST analysis, even with its limitations, compared to technologies such as whole-genome sequencing, contributes to the understanding of genetic diversity, and in our case, to the relationship with the acquisition and dissemination of resistance mechanisms in *A. baumannii*, making the epidemiological surveillance of this type of isolates indispensable. The epidemiological surveillance of Gram-negative ESKAPE MDR bacteria in the hospital environment is an immediate necessity, since it has been observed that, in times of contingency, the emergence of MDR pathogens is drastically increased. Undoubtedly, since high-risk STs are microorganisms defined as those with high genomic plasticity, the possibilities of acquiring and maintaining stable resistance mechanisms are also high.

## 5. Conclusions

From the evidence showed in the present work, Gram-negative ESKAPE bacteria and in particular *A. baumannii* have had an important role during the COVID-19 pandemic in HJM since they have shown a significant increase in resistance to antibiotics. On the other hand, the epidemiological surveillance of antimicrobial resistance in pathogens that cause HAIs is of the utmost importance for hospital centers since, derived from these findings, measures can be taken to control and contain pathogens that, according to the WHO, have been categorized according to critical priority.

## Figures and Tables

**Figure 1 pathogens-13-00050-f001:**
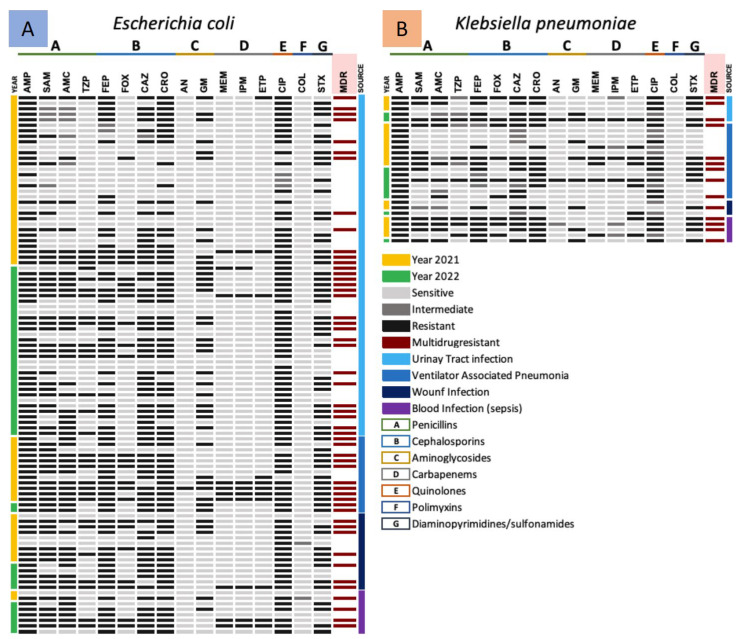
Heat map of the antimicrobial resistance profiles of the *Enterobacterales* group (*E. coli* and *K. pneumoniae* strains) of the Gram-negative ESKAPE bacteria by isolation source and year (2021 and 2022). (**A**) *Escherichia coli* and (**B**) *Klebsiella pneumoniae* strains.

**Figure 2 pathogens-13-00050-f002:**
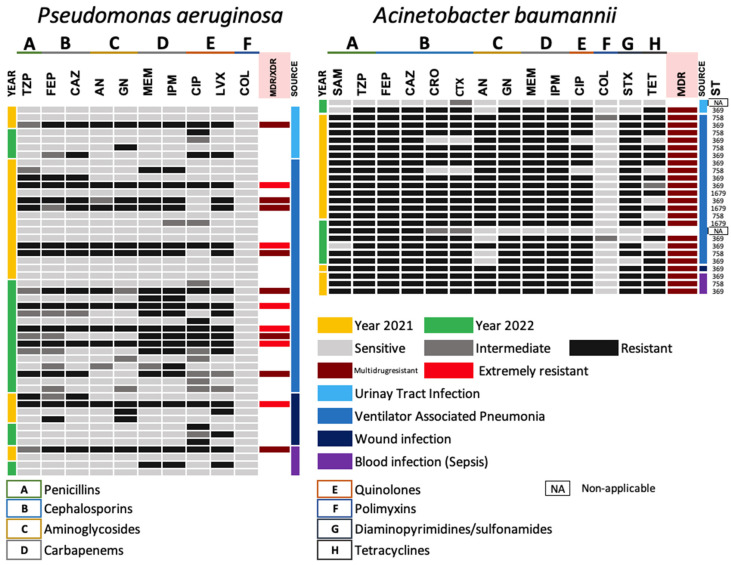
Heat map of the antimicrobial resistance profiles of the Gram-negative ESKAPE bacteria non-lactose fermenting by isolation source and year (2021 and 2022). (**left**) *Pseudomonas aeruginosa* and (**right**) *Acinetobacter baumannii* strains.

**Figure 3 pathogens-13-00050-f003:**
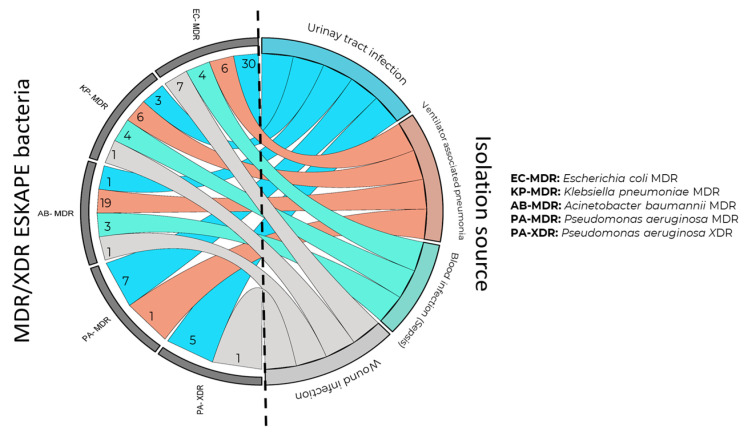
Distribution analysis of MDR and XDR phenotypes of Gram-negative ESKAPE bacteria by isolation source from patients of Hospital Juárez de México (2021–2022).

**Figure 4 pathogens-13-00050-f004:**
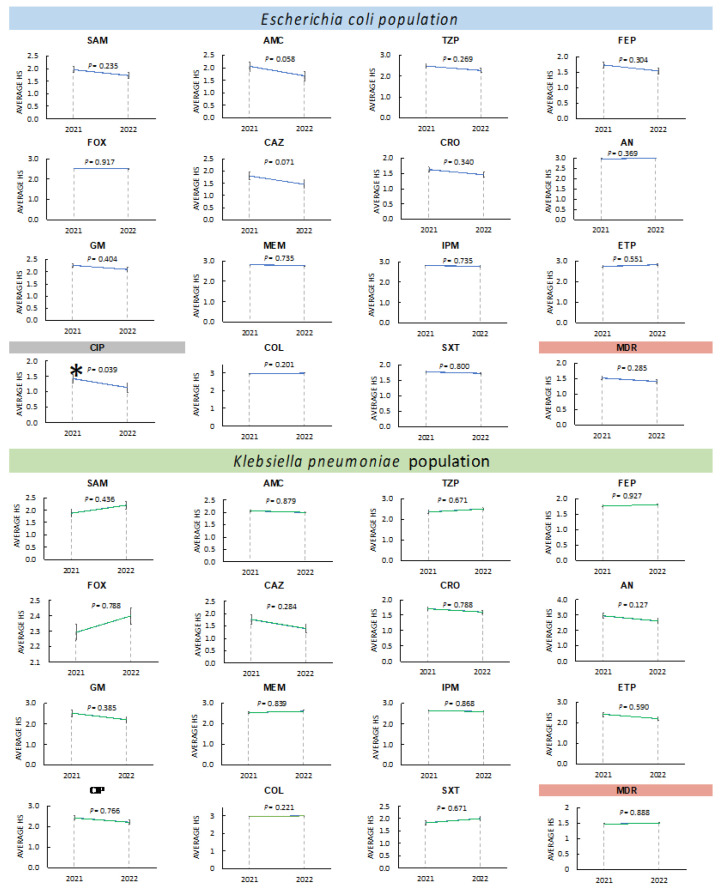
ANOVA analysis and Tukey’s test for the identification of changes in the resistant phenotypes of the group of *Enterobacterales* of Gram-negative ESKAPE bacteria. *Escherichia coli* and *Klebsiella pneumoniae* population. Antimicrobial abbreviations: SAM, ampicillin/sulbactam; AMC, amoxicillin/Ac. clavulanic; TZP, piperacillin/tazobactam; FEP, cefepime; FOX, cefoxitin; CAZ, ceftazidime; CRO, ceftriaxone; AN, amikacin; GM, gentamicin; MEM, meropenem; IPM, imipenem; ETP, ertapenem; CIP, ciprofloxacin; COL, colistin; STX, trimethoprim/sulfamethoxazole. * Significant difference (*p* = 0.05).

**Figure 5 pathogens-13-00050-f005:**
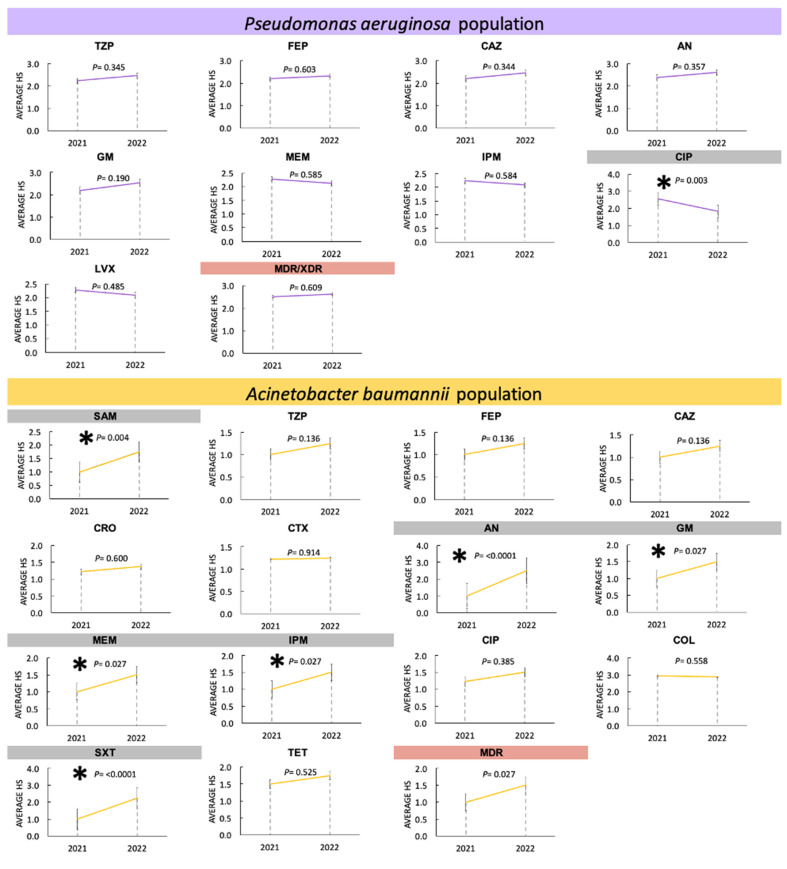
ANOVA analysis and Tukey’s test for the identification of changes in the resistant phenotypes of the group of *Enterobacterales* of Gram-negative ESKAPE bacteria. *Pseudomonas aeruginosa* and *Acinetobacter baumannii* population. Antimicrobial abbreviations: SAM, ampicillin/sulbactam; TZP, piperacillin/tazobactam; FEP, cefepime; CAZ, ceftazidime; CRO, ceftriaxone; CTX, cefotaxime; AN, amikacin; GM, gentamicin; MEM, meropenem; IPM, imipenem; CIP, ciprofloxacin; LVX, levofloxacin; COL, colistin; STX, trimethoprim/sulfamethoxazole. * Significant difference (*p* = 0.05).

**Table 1 pathogens-13-00050-t001:** Primers used in this study.

Primer	Molecular Target	Sequence (5′→3′)	Size (bp)	Reference
*IMP-F*	*bla_IMP_*	TTGACACTCCATTTACDG	139	[35]
*IMP-R*		GATYGAGAATTAAGCCACYCT		
*VIM-F*	*bla_VIM_*	GATGGTGTTTGGTCGCATA	390	
*VIM-R*		CGAATGCGCAGCACCAG		
*KPC-F*	*bla_KPC_*	CATTCAAGGGCTTTCTTGCTGC	538	
*KPC-R*		ACGACGGCATAGTCATTTGC		
*OXA-48F*	*bla_OXA-48_*	GCACTTCTTTTGTGATGGC	281	
*OXA-48R*		GAGCACTTCTTTTGTGATGGC		
*NDM-F*	*bla_NDM_*	GGTTTGGCGAT CTGGTTTTC	621	[36]
*NDM-R*		CGGAATGGCTCATCACGATC		
*OXA-23F*	*bla_OXA-23_*	GATGTGTCATAGTATTCGTCG	1065	[37]
*OXA-23R*		TCACAACAACTAAAAGCACTG		
*OXA-40F*	*bla_OXA-40_*	TCTAGTTTCTCTCAGTGCATGTTCATC	749	[38]
*OXA-40R*		CATTACGAATAGAACCAGACATTCC		
*gltA-F*	Citrate synthase	AATTTACAGTGGCACATTAGGTCCC	722	[39]
*gltA-R*		GCAGAGATACCAGCAGAGATACACG		
*gyrB-F*	DNA gyrase subunit B	TGTAAAACGACGGCCAGTGCNGGRTCYTTYTCYTGRCA	909	
*gyrB-R*		CAGGAAACAGCTATGACCAYGSNGGNGGNAARTTYRA		
*gdhB-F*	Glucose dehydrogenase B	GCTACTTTTATGCAACAGAGCC	775	
*gdhB-R*		GTTGAGTTGGCGTATGTTGTGC		
*recA-F*	Homologous recombination factor	CCTGAATCTTCYGGTAAAAC	425	
*recA-R*		GTTTCTGGGCTGCCAAACATTAC		
*cpn60-F*	60 kDa chaperonin	ACTGTACTTGCTCAAGC	479	
*cpn60-R*		TTCAGCGATGATAAGAAGTGG		
*gpi-F*	Glucose-6-phosphate isomerase	AATACCGTGGTGCTACGGG	508	
*gpi-R*		AACTTGATTTTCAGGAGC		
*rpoD-F*	RNA polymerase sigma factor *rpoD* (Sigma-70)	ACGACTGACCCGGTACGCATGTAYATGMGNGARATCGC NACNCT	492	
*rpoD-R*		ATAGAAATAACCAGACGTAAGTTNGCYTCNACCATYTG YTTYTT		

**Table 2 pathogens-13-00050-t002:** Gram-negative ESKAPE strains analyzed causing nosocomial infections in 2021 and 2022 years from patients of Hospital Juárez de México.

Gram-Negative ESKAPE Bacteria Causing Nosocomial Infections	Analyzed Strains by Year *n* (%)		Total *n* (%)
	2021	2022	
*Escherichia coli*	54 (47.4)	44 (51.2)	98 (49.0)
*Klebsiella pneumoniae*	17 (14.9)	10 (11.6)	27 (13.5)
*Pseudomonas aeruginosa*	25 (21.9)	24 (27.9)	49 (24.5)
*Acinetobacter baumannii*	18 (15.8)	8 (9.30)	26 (13.0)
Total	114 (100)	86 (100)	200 (100)

**Table 3 pathogens-13-00050-t003:** Clinical origin by nosocomial infection of Gram-negative ESKAPE strains analyzed in the years 2021 and 2022 from patients of Hospital Juárez de México.

Clinical Origin	Analyzed Strains by Year *n* (%)		Total
	2021	2022	
Urinary tract infection (UTI)	37 (32.4)	39 (45.3)	76 (38.0)
Ventilator-associated pneumonia (VAP)	50 (43.9)	29 (33.7)	79 (39.5)
Wound infection	16 (14.0)	9 (10.5)	25 (12.5)
Blood infection (Sepsis)	11 (9.7)	9 (10.5)	20 (10.0)
Total	114 (100)	86 (100)	200 (100)

**Table 4 pathogens-13-00050-t004:** Stages in the detection of carbapenemases and the associated resistance mechanisms (included efflux pump *adeABCRS*) in the members of the Gram-negative ESKAPE group of HJM.

ESKAPE	Carbapenemase Detection (*n*/%)									Efflux Pump *adeABC*
	Phenotype *n* (%) by		Genotype by End-Point PCR							
	Disc Diffusion	mCIM Assay	*bla_NDM_*	*bla_VIM_*	*bla_IMP_*	*bla_KPC_*	*bla_OXA-48_*	*bla_OXA-40_*	*bla_OXA-23_*	
*E. coli*	12 (17.6)	8 (25.85)	4 (50)	2 (25)	0 (0)	1 (12.5)	1 (12.5)	NA*	NA	NA
*K. pneumoniae*	11 (16.2)	4 (12.9)	1 (25)	1 (25)	0 (0)	0 (0)	2 (50)	NA	NA	NA
*P. aeruginosa*	21 (30.8)	2 (6.5)	0 (0)	2 (100)	0 (0)	0 (0)	0 (0)	NA	NA	NA
*A. baumannii*	24 (35.4)	17 (54.8)	0 (0)	0 (0)	0 (0)	0 (0)	0 (0)	17 (100)	0 (0)	24 (100)
Total	68 (100)	31 (100)								

NA*: Non-applicable.

## Data Availability

Bello-López, Juan Manuel (2024), “Gram-Negative ESKAPE Bacteria Surveillance in COVID-19 Pandemic Exposes High-Risk Sequence Types of Acinetobacter baumannii MDR in a Tertiary Care Hospital”, Mendeley Data, V1, doi: 10.17632/cyfzc9hnxw.1.

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
