# Peer review of "Gram-Negative ESKAPE Bacteria Surveillance in COVID-19 Pandemic Exposes High-Risk Sequence Types of Acinetobacter baumannii MDR in a Tertiary Care Hospital"

_pathogens, 2024, doi:10.3390/pathogens13010050_

Round 1
Reviewer 1 Report
Comments and Suggestions for Authors
Dear Authors,
The Manuscript shows the potential for an expanding the understanding of the molecular epidemiology of ESCAPE pathogens during the COVID-19 pandemic. However, some items raise concerns and necessitate a revision before the acceptance for publication. Please find below the detailed comments and suggestions.
The article lacks information about the dynamics of nosocomial infections in the hospital. It is unclear, as a result, whether the increase in antibiotic resistance you noted in Acinetobacter was the result of an local outbreak with the spread of multi-resistant clones or increasing in admission of infected patients to the hospital during pandemic?
A note to the title : to avoid misunderstanding, I suggest change « high-risk Type Sequences» to «high-risk sequence type»
Table 4. Please place the table under its heading.
Author Response
Manuscript pathogens-2759108:
ESKAPE bacteria surveillance in COVID-19 pandemic, exposes high-risk Type Sequences of Acinetobacter baumannii MDR in a tertiary care hospital.
Responses to Reviewer 1
We appreciate the reviewer's thorough review of our manuscript, her/his thoughtful comments, and suggestions. We decided to revise our workand to answer point-by-point the reviewer’s queries and made the requested changes to our manuscript.
Comment 1: The article lacks information about the dynamics of nosocomial infections in the hospital. It is unclear, as a result, whether the increase in antibiotic resistance you noted in Acinetobacter was the result of a local outbreak with the spread of multi-resistant clones or increasing in admission of infected patients to the hospital during pandemic?
Answer to comment 1: Dear reviewer, we appreciate your valuable comment. We have made some changes to the manuscript that will clarify his observations. The changes were the following:
- In “5 Detection of high-risk type sequences in Acinetobacter baumannii” section, the text was included as follow:
“According to the identification of the high-risk sequences type during the study period, no outbreaks were identified among patients (Figure 2).”
- Figure 2B (Heat map of A. baumannii resistance phenotypes) was modified, indicating the high-risk sequences type during the study period (2021 and 2022).
- In discussion section, the text: “This suggests the emergence of new carbapenem-resistant strains of baumannii and endowed with other resistance mechanisms, such as efflux pumps previously detected [5,7]. Was modified as follow: “This suggests the emergence of new carbapenem-resistant strains of A. baumannii and endowed with other resistance mechanisms, such as efflux pumps previously detected and consequently confer resistance to a wide variety of antibiotics, including those where a significant increase was identified [5,7].
Comment 2: A note to the title: to avoid misunderstanding, I suggest change « high-risk Type Sequences» to «high-risk sequence type»
Answer to comment 2: Dear reviewer, we appreciate your valuable comment. We made the change to the entire manuscript (including the title of the manuscript).
Comment 3: Table 4. Please place the table under its heading.
Answer to comment 3: Dear reviewer, we appreciate your valuable comment. We made the suggested change.

Reviewer 2 Report
Comments and Suggestions for Authors
This study monitored the prevalence and antibiotic resistance of ESKAPE bacteria during the COVID-19 pandemic.
1. The study lacks novelty. It is an extension report of previous research. The team had previously published findings before and during the first year of the COVID-19 pandemic. This report covers the results from the second and third years.
2. In the introduction, the author introduced ESKAPE and explained different definitions. However, the author did not specify how ESKAPE was defined in this study. If following a general definition, the author needs to explain why the study excluded Enterococcus faecium and Staphylococcus aureus. Even if "E" is defined as Enterobacterales and "S" as Stenotrophomonas maltophilia, there still needs to be an explanation for not encompassing all species in the research.
3. There are several layout errors in the article, with discrepancies between the positioning of table titles and figure legends and their respective tables and figures.
4. It is necessary to provide a more detailed explanation of the strain collection strategy, such as whether all strains are included or if they are randomly selected.
5. MEM should not be classified as an aminoglycoside (Line 246).
6. The figure depicting antibiotic resistance changes (Figure 4 and 5) could be consolidated into a single figure or presented in a table format.
7. In lines 313-317, the author initially explains that among 15 antibiotics tested in E. coli and K. pneumoniae, 14 did not show significant changes in resistance, then specifies that ciprofloxacin exhibited significant changes. However, it should be clarified that significant changes in resistance to ciprofloxacin were observed only in E. coli, which might cause confusion in the author's statement.
8. The figure legend for Figure 4 appears twice, whereas Figure 5 lacks a figure legend.
9. In lines 426-427, the author wrote, "...isolation rates increased dramatically..., from 23 and 17% for P. aeruginosa and A. baumannii in 96 cases...," suggesting that the percentage of increase should be explicitly stated.
Author Response
Manuscript pathogens-2759108:
ESKAPE bacteria surveillance in COVID-19 pandemic, exposes high-risk Type Sequences of Acinetobacter baumannii MDR in a tertiary care hospital.
Responses to Reviewer 2
We appreciate the reviewer's thorough review of our manuscript, her/his thoughtful comments, and suggestions. We decided to revise our workand to answer point-by-point the reviewer’s queries and made the requested changes to our manuscript.
Comment 1: The study lacks novelty. It is an extension report of previous research. The team had previously published findings before and during the first year of the COVID-19 pandemic. This report covers the results from the second and third years.
Answer to comment 1: Dear reviewer, we appreciate your valuable comment. Effectively, it is a study to surveillance the behavior of antimicrobial resistance in gram-negative bacteria from the ESKAPE group of our hospital. We consider the document provides sufficient information to demonstrate the change in antimicrobial resistance patterns in the main microorganism involved in HAIS (Acinetobacter baumannii), along with the genetic evidence of the turnover of carbapenem resistance markers in this microorganism.
Comment 2: In the introduction, the author introduced ESKAPE and explained different definitions. However, the author did not specify how ESKAPE was defined in this study. If following a general definition, the author needs to explain why the study excluded Enterococcus faecium and Staphylococcus aureus. Even if "E" is defined as Enterobacterales and "S" as Stenotrophomonas maltophilia, there still needs to be an explanation for not encompassing all species in the research.
Answer to comment 2: Dear reviewer, we appreciate your valuable comment.
In Line 92 (Introduction section) the following text:
“In Mexico, we have reported locally the main members of the ESKAPE group resistant to antibiotics circulating in pre-pandemic and pandemic times in patients of the Hospital Juárez de México (HJM) [7,14], as well as the consequences of their dispersion on inert surfaces and critical devices [5,22-25].”
Was modified as follow:
“In Mexico, we have reported locally the members of the gram-negative ESKAPE group resistant to antibiotics (mainly gram-negative bacteria) circulating in pre-pandemic and pandemic times in patients of the Hospital Juárez de México (HJM) [7,14], as well as the consequences of their dispersion on inert surfaces and critical devices such as A. baumannii [5,22-25]. Therefore, microbiological evidence from our hospital has shown that only some gram-negative bacteria (A. baumannii, P. aeruginosa and Enterobacterales) have played an important role as causal agents of HAIS and nosocomial contamination”.
In line 127, the following text was included as follow:
“Because gram-negative bacteria from the gram-negative ESKAPE group have been the most prevalent as causal agents of HAIS in HJM, only these were included in the study.”
Finally, under this observation, we additionally made modifications throughout the manuscript to clarify that our research was focused only on gram-negative bacteria from the ESKAPE group.
Comment 3: There are several layout errors in the article, with discrepancies between the positioning of table titles and figure legends and their respective tables and figures.
Answer to comment 3: Dear reviewer, we appreciate your valuable comment. We have corrected the distribution of figures, tables and titles of figures and tables.
Comment 4: It is necessary to provide a more detailed explanation of the strain collection strategy, such as whether all strains are included or if they are randomly selected.
Answer to comment 4: Dear reviewer, we appreciate your valuable comment. The text:
“The gram-negative ESKAPE strains (Escherichia coli, Klebsiella pneumoniae, Pseudomonas aeruginosa, and Acinetobacter baumannii) were isolated from the non-repeat patients from the HJM, during 2021 (n=114) and 2022 (n=86).”
Was modified as follow:
“The gram-negative ESKAPE strains (Escherichia coli, Klebsiella pneumoniae, Pseudomonas aeruginosa, and Acinetobacter baumannii) were isolated from the non-repeat patients coursing confirmed HAIS (without previous antimicrobial treatment) from the HJM, during 2021 (n=114) and 2022 (n=86).”
Comment 5: MEM should not be classified as an aminoglycoside (Line 246).
Answer to comment 5: Dear reviewer, we appreciate your valuable comment. The text: “and MEM/22%” was deleted.
Comment 6: The figure depicting antibiotic resistance changes (Figure 4 and 5) could be consolidated into a single figure or presented in a table format.
Answer to comment 6: Dear reviewer, we appreciate your valuable comment. We have done a lot of redesigning tests with Figures 4 and 5, but unfortunately it loses resolution when trying to place it on a single page. We consider that they are very valuable visual elements to show the main aim of the work, however, if you know of any way to show the two figures into one, we would appreciate your feedback.
On the other hand, his valuable observation served to detect an error in Figure 4 (the p value for ciprofloxacin in the K. pneumoniae population). This error has already been corrected and does not affect the content of the manuscript.
Comment 7: In lines 313-317, the author initially explains that among 15 antibiotics tested in E. coli and K. pneumoniae, 14 did not show significant changes in resistance, then specifies that ciprofloxacin exhibited significant changes. However, it should be clarified that significant changes in resistance to ciprofloxacin were observed only in E. coli, which might cause confusion in the author's statement.
Answer to comment 7: Querido revisor, agradecemos su valioso comentario. Derivado de su observación, realizamos los siguientes cambios en la sección de resultados (3.3.3) as follow:
The subtitle “3.3.3 No significant changes in resistance in Enterobacterales during 2021 and 2022”, was changed by “3.3.3 Changes in antimicrobial resistance for Enterobacterales during 2021 and 2022.”
The subtitle “3.3.4 Significant changes in antimicrobial resistance in non-fermenters during 2021 and 2022” was changed by “3.3.4 Changes in antimicrobial resistance in non-fermenters during 2021 and 2022.“
Finally, the following text (in 3.3.3 section):
“The results of the analysis revealed that there was no significant difference (p=0.05) between the means of the results of the resistant phenotypes obtained from the E. coli and K. pneumoniae profiles in 14 of the 15 antibiotics tested. A significant difference in resistance (with a downward trend in 2022) was only identified for ciprofloxacin (p=0.039). Finally, no significant variation was detected in the identification of MDR isolates in the Enterobacteriaceae in the study period. Figure 4 shows the analysis of variance and Tukey’s test for the identification of changes in the resistant phenotypes of the group of Enterobacterales of ESKAPE bacteria from the HJM.”
Was modified as follow:
The results of the analysis revealed that there was no significant difference (p=0.05) between the means of the results of the resistant phenotypes obtained from the E. coli profiles in 14 of the 15 antibiotics tested. A significant difference in resistance (with a downward trend in 2022) was only identified for ciprofloxacin (p=0.039). Finally, no significant variation was detected in K. pneumoniae population and global identification of MDR isolates in the Enterobacteriales. Figure 4 shows the analysis of variance and Tukey’s test for the identification of changes in the resistant phenotypes of the group of Enterobacterales of gram-negative ESKAPE bacteria from the HJM.
Comment 8: The figure legend for Figure 4 appears twice, whereas Figure 5 lacks a figure legend.
Answer to comment 8: Dear reviewer, we appreciate your valuable comment. The legends of figures 4 and 5 were in the manuscript but due to the conversion to PDF they were modified. That error has already been corrected.
Comment 9: In lines 426-427, the author wrote, "...isolation rates increased dramatically..., from 23 and 17% for P. aeruginosa and A. baumannii in 96 cases...," suggesting that the percentage of increase should be explicitly stated.
Answer to comment 9: Dear reviewer, we appreciate your valuable comment. Derived from his observation, the following text:
“In contrast, during the first year of the pandemic, isolation rates increased dramatically in VAP cases in COVID-19 patients, from 23 and 17% for P. aeruginosa and A. baumannii in 96 cases in only two months of the first pandemic wave [7].”
Was modified as follow:
“In contrast, during the first year of the pandemic, isolation rates increased dramatically in VAP cases in COVID-19 patients, from 20% (n=19) and 15% (n=14) for P. aeruginosa and A. baumannii in 36 cases in only two months of the first pandemic wave [7].”

Reviewer 3 Report
Comments and Suggestions for Authors
The authors have described the ESKAPE pathogens isolated from patients at Hospital Juárez de México from four different clinical samples (wound, sepsis, UTI or VAP). This is a common area of research to understand the sequence types of ESKAPE pathogens and their antimicrobial resistance characteristics which contributes to overall knowledge on the pathogens in the different parts of the world. This is the first time that someone has described the ESKAPE pathogens and antimicrobial resistance characteristics in this hospital.
Overall, the paper is a well put together study on the surveillance of ESKAPE pathogens in the COVID-19 pandemic. I recommend the authors review their manuscript to ensure all the figures and tables are aligned with their accompanying titles on the same page with page breaks where necessary. This would help the reviewers to correlate the missing figures referred to in the results section.
Since sequencing platforms are available at reduced cost these days, authors could perform phylogenetic tree analysis of the strains and look at the STs dominating in the cohort and understand the presence of nosocomial spread (if any) in the hospital population since they do not look at non-repeat patients.
The paper lacks a good conclusion section and the discussion seems to be a review of different papers so it would be helpful if the authors could provide a conclusion based on their results and their research question.
More references would help support the claims of the authors. For example, in the introduction lines 64-66, the study cited has limited sample sets and hence would be helpful to include a reference with a larger sample set. In addition, around four of the references seem to be from previous publications.
NA
Author Response
Manuscript pathogens-2759108:
ESKAPE bacteria surveillance in COVID-19 pandemic, exposes high-risk Type Sequences of Acinetobacter baumannii MDR in a tertiary care hospital.
Responses to Reviewer 3
We appreciate the reviewer's thorough review of our manuscript, her/his thoughtful comments, and suggestions. We decided to revise our workand to answer point-by-point the reviewer’s queries and made the requested changes to our manuscript.
Reviewer 3
The authors have described the ESKAPE pathogens isolated from patients at Hospital Juárez de México from four different clinical samples (wound, sepsis, UTI or VAP). This is a common area of research to understand the sequence types of ESKAPE pathogens and their antimicrobial resistance characteristics which contributes to overall knowledge on the pathogens in the different parts of the world. This is the first time that someone has described the ESKAPE pathogens and antimicrobial resistance characteristics in this hospital.
Comment 1:
Overall, the paper is a well put together study on the surveillance of ESKAPE pathogens in the COVID-19 pandemic. I recommend the authors review their manuscript to ensure all the figures and tables are aligned with their accompanying titles on the same page with page breaks where necessary. This would help the reviewers to correlate the missing figures referred to in the results section.
Answer to comment 1: Dear reviewer, we appreciate your valuable comment. A thorough review of the manuscript has been carried out and we have corrected errors in the alignment of tables and figures, as well as reviewing all references.
Comment 2: Since sequencing platforms are available at reduced cost these days, authors could perform phylogenetic tree analysis of the strains and look at the STs dominating in the cohort and understand the presence of nosocomial spread (if any) in the hospital population since they do not look at non-repeat patients.
Answer to comment 2: Dear reviewer, your comment is definitely perfectly grounded in reality. We are currently developing a research project based on the sequencing and assemblies of the Acinetobacter baumannii genomes. This work will be part of a manuscript that we will present for publication in the future.
Comment 3: The paper lacks a good conclusion section, and the discussion seems to be a review of different papers so it would be helpful if the authors could provide a conclusion based on their results and their research question.
Answer to comment 3: Dear reviewer, we appreciate your valuable comment. A section of conclusions has been included, hoping for your approval because we adhere to the results and proposed research question. The generated section is the following:
- Conclusions
From the evidence showed in the present work, gram-negative ESKAPE bacteria and in particular A. baumannii have and have had an important role during the COVID-19 pandemic in the HJM since they have shown a significant increase in resistance to antibiotics. On the other hand, epidemiological surveillance of antimicrobial resistance in pathogens that cause HAIS is of utmost importance for hospital centers since, derived from these findings, measures can be taken to control and contain pathogens that, according to the WHO, have been categorized according to critical priority.
Comment 4: More references would help support the claims of the authors. For example, in the introduction lines 64-66, the study cited has limited sample sets and hence would be helpful to include a reference with a larger sample set. In addition, around four of the references seem to be from previous publications.
Answer to comment 4: Dear reviewer, we appreciate your valuable comment. Derived from his comment, we decided to include three references [15,16 and 17] that enrich the argument indicated in line 64-66. The references are the following:
- Kaier, K.; Heister, T.; Motschall, E.; Hehn, P.; Bluhmki, T.; Wolkewitz, M. Impact of mechanical ventilation on the daily costs of ICU care: a systematic review and meta regression. Epidemiol Infect. 2019, 147, e314.
- Luckraz, H.; Manga, N.; Senanayake, E.L.; Abdelaziz, M.; Gopal, S.; Charman, S.C.; Giri, R.; Oppong, R.; Andronis, L. Cost of treating ventilator-associated pneumonia post cardiac surgery in the National Health Service: Results from a propensity-matched cohort study. J Intensive Care Soc. 2018,19,94-100.
- Ladbrook, E.; Khaw, D.; Bouchoucha, S.; Hutchinson, A. A systematic scoping review of the cost-impact of ventilator-associated pneumonia (VAP) intervention bundles in intensive care. Am J Infect Control. 2021,7,928-936.
These references support the following text that was added to line 71-73:
“In other regions of the world, the negative impact in terms of costs of HAIS has been demonstrated, where VAPs are considered infectious complications that dramatically increase care costs and mortality [15-17]”.

Reviewer 4 Report
Comments and Suggestions for Authors
Thank you, authors, for the study. Some suggestions for consideration.
is MDR defined as 3 or more antibiotic classes? is this based on phenotypes or genotypes?
could some of the average HS changes between the two year could be due to the difference in proportion of specific site infections and/or specific types of bacteria among ESKAPE which would have secondary effect on the proportion of resistant rate against antibiotics?
the increase of drug resistance during/after COVID-19 pandemic, line 416 could it be due to unnecessary/over-/misuse of antibiotics during COVID-19 as another plausible reason? How about due to seeing more patients admitted leading to relatively more nosocomial infections as compared to other period with relatively less patient admission?
Methods: were these patients treated with antibiotics prior to the isolation of ESKAPE from their biological specimens?
Figure 3 and 4: suggest to indicate * for p<0.05 for readers' ease of reference
Line 374: were there any patterns in terms of antibiotic resistance profile (phenotype/genotype) between the three high-risk STs of A. baumannii?
Line 417: while agree with authors that microbiological/AMR surveillance would be a key pillar in infection control, how would the reduction of microbiological surveillance be a potential (direct) 'cause' for the increase of drug resistance in these pathogens.
Line 526: were those isolates/pathogens belonged to the same STs epidemiological related e.g. from the same hospital rooms/exposed to same operation theatre during similar period?
Author Response
Manuscript pathogens-2759108:
ESKAPE bacteria surveillance in COVID-19 pandemic, exposes high-risk Type Sequences of Acinetobacter baumannii MDR in a tertiary care hospital.
Responses to Reviewer 4
We appreciate the reviewer's thorough review of our manuscript, her/his thoughtful comments, and suggestions. We decided to revise our workand to answer point-by-point the reviewer’s queries and made the requested changes to our manuscript.
Comment 1: is MDR defined as 3 or more antibiotic classes? is this based on phenotypes or genotypes?
Answer to comment 1: Dear reviewer, we appreciate your valuable comment. The MDR classification is based on two criteria (exclusively PHENOTYPIC): type of microorganism and number of classes of antibiotics that the strain can resist. In the case of Acinetobacter baumannii, its MDR classification is based on resistance to 3 of 11 classes of antibiotics. This criterion also applies to Klebsiella pneumoniae/Escherichia coli, and Pseudomonas aeruginosa. Because the information on categorization is extensive, we decided to refer to the “Latin American consensus to define, categorize and notify multidrug-resistant pathogens, with widespread resistance or pan-resistant pathogens” [29].”
Comment 2: could some of the average HS changes between the two year could be due to the difference in proportion of specific site infections and/or specific types of bacteria among ESKAPE which would have secondary effect on the proportion of resistant rate against antibiotics?
Answer to comment 2: Dear reviewer, we appreciate your valuable comment. Indeed, the difference in the proportion of infections (by type of infection and infectious agent) could modify the average HS. However, regardless of the variation, the ANOVA test coupled with Tukey's test allows us to validate whether there is a significant difference between the susceptibility profiles in the years evaluated.
Comment 3: The increase of drug resistance during/after COVID-19 pandemic, line 416 could it be due to unnecessary/over-/misuse of antibiotics during COVID-19 as another plausible reason? How about due to seeing more patients admitted leading to relatively more nosocomial infections as compared to other period with relatively less patient admission?
Answer to comment 3: Dear reviewer, we appreciate your valuable comment. Derived from his comment we have modified the following paragraph:
Theories such as the increase of patients with comorbidities in the COVID-19 period, and reduction of microbiological surveillance were recognized as potential causes for the increase of drug resistance in these pathogens.
Was modified as follow:
Theories such as the increase of patients with comorbidities in the COVID-19 period, reduction of microbiological surveillance, increase in the population treated in hospitals and unnecessary, irrational and misuse of antibiotics were recognized as potential causes for the increase of drug resistance in these pathogens.
Comment 4: Methods: were these patients treated with antibiotics prior to the isolation of ESKAPE from their biological specimens?
Answer to comment 4: Dear reviewer, we appreciate your valuable comment. In Methods section the follow text:
“The gram-negative ESKAPE strains (Escherichia coli, Klebsiella pneumoniae, Pseudomonas aeruginosa, and Acinetobacter baumannii) were isolated from the non-repeat patients from the HJM, during 2021 (n=114) and 2022 (n=86).”
Was modified as follow:
“The gram-negative ESKAPE strains (Escherichia coli, Klebsiella pneumoniae, Pseudomonas aeruginosa, and Acinetobacter baumannii) were isolated from the non-repeat patients coursing confirmed HAIS (without previous antimicrobial treatment) from the HJM, during 2021 (n=114) and 2022 (n=86).”
Comment 5: Figure 3 and 4: suggest to indicate * for p<0.05 for readers' ease of reference.
Answer to comment 5: Dear reviewer, we appreciate your valuable comment. We modified the Figure 3 and 4.
Comment 6: Line 374: were there any patterns in terms of antibiotic resistance profile (phenotype/genotype) between the three high-risk STs of A. baumannii?
Answer to comment 6: Dear reviewer, we appreciate your valuable comment. As you can see in Table 4, the population of A. baumannii (where the three high-risk STs are) were carriers of the gene encoding a carbapenemase blaOXA-40 (WITH PHENOTYPICAL PRODUCTION OF CARBAPENEMASES). The above can be seen in table 4. Additionally, in section 3.5 the finding is mentioned as follows:
“All the high-risk clones identified carried the gene encoding blaOXA-40 and the efflux pump AdeABCRS”
Comment 7: Line 417: while agree with authors that microbiological/AMR surveillance would be a key pillar in infection control, how would the reduction of microbiological surveillance be a potential (direct) 'cause' for the increase of drug resistance in these pathogens.
Answer to comment 7: Dear reviewer, we appreciate your valuable comment. In relation to his comment, the following information was added that clarifies the neglect in microbiological surveillance.
The text:
“Theories such as the increase of patients with comorbidities in the COVID-19 period, and reduction of microbiological surveillance were recognized as potential causes for the increase of drug resistance in these pathogens.”
Were modified as follow:
“Theories such as the increase of patients with comorbidities in the COVID-19 period, and reduction of microbiological surveillance (report of microbiological contamination of surfaces, critical and semi-critical devices, report to antimicrobial control committees, infection control committees, among others.), increase in the population treated in hospitals and unnecessary, irrational and misuse of antibiotics were recognized as potential causes for the increase of drug resistance in these pathogens.”
Comment 8: Line 526: were those isolates/pathogens belonged to the same STs epidemiological related e.g. from the same hospital rooms/exposed to same operation theatre during similar period?
Answer to comment 8: Dear reviewer, we appreciate your valuable comment. We decided to include in section 3.5, text that clarifies its observation. We also decided to modify Figure 2, including the assignment of sequence type for each Acinetobacter baumannii isolate.
“According to the identification of the high-risk Sequences Type during the study period, no outbreaks were identified among patients (Figure 2B). All the high-risk clones identified carried the gene encoding blaOXA-40 and the efflux pump AdeABCRS”.

Round 2
Reviewer 2 Report
Comments and Suggestions for Authors
All questions have been replied to.